# DYNAMIC PLANNING NETWORKS

## ABSTRACT

We introduce Dynamic Planning Networks (DPN), a novel architecture for deep reinforcement learning, that combines model-based and model-free aspects for online planning. Our architecture learns to dynamically construct plans using a learned state-transition model by selecting and traversing between simulated states and actions to maximize valuable information before acting. In contrast to model-free methods, model-based planning lets the agent efficiently test action hypotheses without performing costly trial-and-error in the environment. DPN learns to efficiently form plans by expanding a single action-conditional state transition at a time instead of exhaustively evaluating each action, reducing the required number of state-transitions during planning by up to 96%. We observe various emergent planning patterns used to solve environments, including classical search methods such as breadth-first and depth-first search. DPN shows improved data efficiency, performance, and generalization to new and unseen domains in comparison to several baselines.

## 1 INTRODUCTION

The central focus of reinforcement learning (RL) is the selection of optimal actions to maximize the expected reward in an environment where the agent must rapidly adapt to new and varied scenarios. Various avenues of research have spent considerable efforts improving core axes of RL algorithms such as performance, stability, and sample efficiency. Significant progress on all fronts has been achieved by developing agents using deep neural networks with model-free RL (Mnih et al., 2015; 2016; Schulman et al., 2015; 2017; OpenAI, 2018); showing model-free methods efficiently scale to high-dimensional state space and complex domains with increased compute. Unfortunately, model-free policies are often unable to generalize to variances within an environment as the agent learns a policy which directly maps environment states to actions.

A favorable approach to improving generalization is to combine an agent with a learned environment model, enabling it to reason about its environment. This approach, referred to as model-based RL learns a model from past experience, where the model usually captures state-transitions, $p(s_{t+1}|s_t, a_t)$, and might also learn reward predictions $p(r_{t+1}|s_t, a_t)$. Usage of learned state-transition models is especially valuable for planning, where the model predicts the outcome of proposed actions, avoiding expensive trial-and-error in the actual environment – improving performance and generalization. This contrasts with model-free methods which are explicitly trial-and-error learners (Sutton & Barto, 2017). Historically, applications have primarily focused on domains where a state-transition model can be easily learned, such as low dimensional observation spaces (Peng & Williams, 1993; Deisenroth & Rasmussen, 2011; Levine & Abbeel, 2014), or where a perfect model was provided (Coulom, 2006; Silver et al., 2016a) – limiting usage. Furthermore, application to environments with complex dynamics and high dimensional observation spaces has proven difficult as state-transition models must learn from agent experience, suffer from compounding function approximation errors, and require significant amounts of samples and compute (Oh et al., 2015; Chiappa et al., 2017; Guzdial et al., 2017). Fortunately, recent work has overcome the aforementioned difficulties by learning to interpret imperfect model predictions (Weber et al., 2017) and learning in a lower dimensional state space (Farquhar et al., 2017a).

Planning in RL has used state-transition models to perform simulated trials with various styles of state traversal such as: recursively expanding all available actions per state for a fixed depth (Farquhar et al., 2017a), expanding all actions of the initial state and simulating forward for a fixed number of steps with a secondary policy (Weber et al., 2017), or performing many simulated roll-

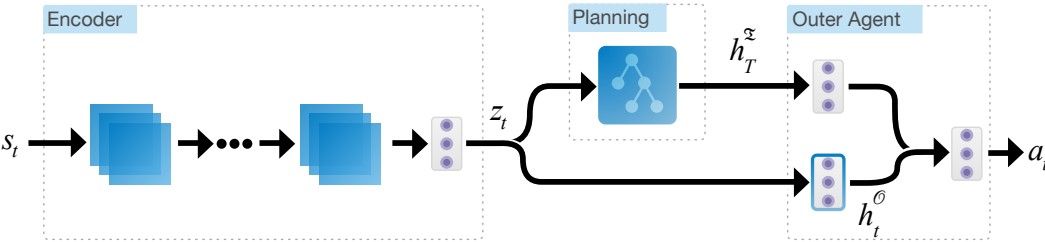

Figure 1: Dynamic Planning Network Architecture. Encoder is comprised of several convolutional layers and a fully-connected layer. Planning occurs for $\tau = 1, ..., T$ steps using the IA and state-transition model. The result of planning is sent to the outer agent before an action $a_t$ is chosen. The fully-connected layer within the outer agent, outlined in blue, is used by the planning process.

outs with each stopping when a terminal state is encountered (Silver et al., 2016a). An issue arises within simulated trials when correcting errors in action selection, as actions can either be undone by wasting a simulation step, using the opposing action, or are irreversible, causing the remaining rollout steps to be sub-optimal in value. Ideally, the agent can step the rollout backwards in time thereby undoing the poor action and choosing a better one in its place. Additionally, during rollouts the agent is forced to either perform a fixed number of steps or continue until a terminal state has been reached; when ideally a rollout can terminate early if the agent decides the path forward is of low value.

In this paper, we propose an architecture that learns to dynamically control a state-transition model of the environment for planning. By doing so, our model has greater flexibility during planning allowing it to efficiently adjust previously simulated actions. We demonstrate improved performance against both model-free and planning baselines on varied environments.

The paper is organized as follows: Section 2 covers our architecture and training procedure, Section 3 covers related work, Section 4 details the experimental design used to evaluate our architecture, and in Section 5 we analyze the experimental results of our architecture.

## 2 DYNAMIC PLANNING NETWORK

In this section, we describe DPN, a novel planning architecture for deep reinforcement learning (DRL). We first discuss the architecture overview followed by the training procedure. Steps taken in the environment use subscript $t$ and steps taken during planning use subscript $\tau$. We provide additional motivation behind the architecture in Appendix A.1.

### 2.1 DPN ARCHITECTURE

The architecture is comprised of an inner agent, an outer agent, a shared encoder, and a learned state-transition model. Figure 1 illustrates a high-level diagram of the DPN architecture. The outer agent (OA) is a feed-forward network and the inner agent (IA) is based on a recurrent neural network (RNN). The architecture interacts with the environment by observing raw environment states $s_t \in \mathcal{S}$ and outputting actions $a_t \in \mathcal{A}$ via OA. However, before OA outputs an action $a_t$ the IA performs $\tau = 1, ..., T$ steps of planning by interacting with an internal simulated environment; where this simulated environment is defined by the state-transition model and sub-section of OA's network. Selection of an action $a_t$ by OA uses the final hidden state $h_T^{\mathcal{I}} \in \mathbb{R}^{1 \times h^i}$ of IA and an embedding of the current state $z_t \in \mathbb{R}^{1 \times z}$. The objective of IA is to maximize the total *utility* provided to the OA; where *utility*, given in Equation 1, measures the "value of information" provided to OA if it were to have undergone a state transition from $z_\tau$ to $z_{\tau+1}$.

$$\mathcal{U}_\tau(h_{\tau+1}^{\mathcal{O}}, h_\tau^{\mathcal{O}}, a_\tau, z_\tau) = \hat{Q}(z_\tau, a_\tau) D_{\mathrm{KL}}(h_{\tau+1}^{\mathcal{O}} || h_\tau^{\mathcal{O}})$$
$$\hat{Q}(z_\tau, a_\tau) = \pi(a_\tau | z_\tau; \theta^{\mathcal{I}}) Q(z_\tau, a_\tau; \theta^{\mathcal{O}}) \tag{1}$$

where $z_{\tau+1}$ is the state transitioned to after performing an action $a_\tau$ in state $z_\tau$, $h_\tau^{\mathcal{O}} \in \mathbb{R}^{1 \times h^o}$ and $h_{\tau+1}^{\mathcal{O}}$ are the hidden states of OA after perceiving the current state $z_\tau$ and state transitioned to $z_{\tau+1}$ respectively, $D_{KL}$ is the KullbackLeibler (KL) distance measure, $Q(z_\tau, a_\tau; \theta^{\mathcal{O}})$ is the value OA assigns to taking action $a_\tau$ in the current state $z_\tau$, and $\pi(a_\tau | z_\tau; \theta^{\mathcal{I}})$ is the learned policy of IA. The $D_{\text{KL}}$ is used to measure the distance in bits between hidden states of the OA. Computation of $D_{\text{KL}}$ between hidden states is provided in the Appendix B.

The *utility* function captures the tension between choosing a state with high value, as described by the Q-function component, and a state that provides maximal information, described by the KL component. To maximize the *utility* the IA provides to the OA it must select states and actions that have both high value and provide information to the OA – maximizing the *"value of information"*. States with high value but low information or low value and high information will have low *utility* and therefore be less desirable.

Therefore, to maximize *utility* for OA during each planning step $\tau$, IA must select appropriate simulated-states $z_\tau^*$ and actions $a_\tau^*$. A simulated-state $z_\tau^*$ is selected from one of three embedded states tracked during planning: the previous $z_\tau^p$, current $z_\tau^c$, and root states $z_\tau^r$; with the triplet written as $z_\tau^{\{p,c,r\}}$ for convenience. Initially, $z_{\tau=0}^{\{p,c,r\}}$ is set to an embedding $z_t$ produced by the encoder of the initial raw state $s_t$ as $z_{\tau=0} = encoder(s_t)$. The encoder is comprised of a series of convolutional layers specific to each environment and provided in the Appendix B. Before planning begins, OA's hidden state is updated using $z_{\tau=0}$:

$$h_\tau^{\mathcal{O}} = W^{zh} z_\tau \tag{2}$$

where $\boldsymbol{W}^{zh} \in \mathbb{R}^{z \times h^o}$ is a learnable parameter of OA with biases omitted. Within this work, we consider the intermediate activation from the OA, a feed-forward network, as a hidden state. The simulated-action $a_\tau^*$ mirrors those available to OA in the environment, such that $a_\tau^* \in \mathcal{A}$.

## 2.2 A PLANNING STEP

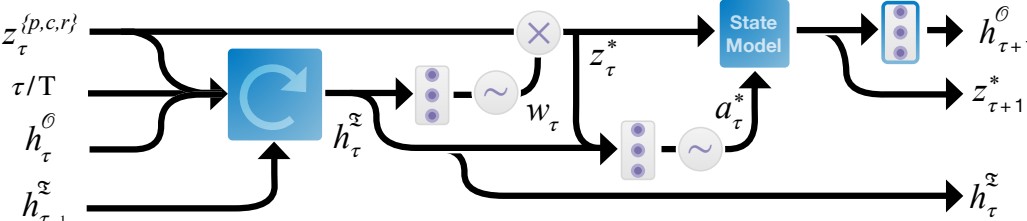

Figure 2: A single planning step $\tau$. Inner Agent, shown as the blue box with a recursive arrow, performs a step of planning using the state-transition model. Circles containing $\times$ indicate multiplication and circles with $\sim$ indicate sampling from the Gumbel Softmax distribution.

At each planning step $\tau$, shown in Figure 2, the IA selects a simulated-state $z_\tau^*$ and action $a_\tau^*$ by considering the previous hidden state $h_{\tau-1}^{\mathcal{I}}$, the triplet of embedded states $z_\tau^{\{p,c,r\}}$, a scalar representing the current planning step $\tau/T$, and OA's hidden state $h_\tau^{\mathcal{O}}$ given $z_\tau^c$. The information is concatenated together forming a context and is fed into IA, a recurrent network producing an updated hidden state $h_\tau^{\mathcal{I}} \in \mathbb{R}^{1 \times h^i}$. The updated hidden state is used to select the simulated-state $z_\tau^*$ by multiplying $z_\tau^{\{p,c,r\}}$ with a 1-hot encoded weight $w_\tau \in \{0,1\}^{1 \times 3}$ sampled from the Gumbel-Softmax distribution, $G$:

$$w_\tau \sim G(\boldsymbol{W}^{h3} h_\tau^{\mathcal{I}})$$
$$z_\tau^* = w_\tau [z_\tau^p, z_\tau^c, z_\tau^r] \tag{3}$$

where $\boldsymbol{W}^{h3} \in \mathbb{R}^{h^i \times 3}$ is a learnable parameter belonging to IA and $G$ is the Gumbel-Softmax distribution (GSD). Where the GSD is a continuous relaxation of the discrete categorical distribution

giving access to differentiable discrete variables (Jang et al., 2016; Maddison et al., 2016). Empirically, we found that using a 1-hot encoding for the weight $w_\tau$ gives greater performance than a softmax activation. Therefore, we used GSD in place of softmax activations throughout our architecture. Next, the simulated-action $a_\tau^* \in \{0, 1\}^{1 \times \mathcal{A}}$, is sampled as follows:

$$a_\tau^* \sim G(\boldsymbol{W}^{azh}[z_\tau^*, h_\tau^{\mathcal{I}}]) \tag{4}$$

where $\boldsymbol{W}^{azh} \in \mathbb{R}^{a \times z + h^i}$ is a learnable parameter of IA. In Equation 4 the selected simulated-state $z_\tau^*$ and IA's hidden state $h_\tau^{\mathcal{I}}$ are concatenated, passed through a linear layer, and used as logits for GSD. Then, with the selected simulated-state $z_\tau^*$ and simulated-action $a_\tau^*$, we produce the next state $z_{\tau+1}$ using the state-transition model, defined as:

$$
\begin{aligned}
z' &= z_\tau + tanh(\boldsymbol{W}^{zz} z_\tau^*) \\
z'' &= z' + tanh((a_\tau^* \boldsymbol{W}^{azz}) z') \\
z_{\tau+1}^* &= z_\tau + z''
\end{aligned}
\tag{5}
$$

where $\boldsymbol{W}^{zz} \in \mathbb{R}^{z \times z}$ and $\boldsymbol{W}^{azz} \in \mathbb{R}^{\mathcal{A} \times z \times z}$ are learnable parameters of the state-transition model. We parameterize each available action in $\mathcal{A}$ with a learned weight matrix that carries information about the effect of taking an action $a_\tau^* \in \mathbb{R}^{1 \times \mathcal{A}}$. We use the same state-transition model presented by Farquhar et al. (2017a). Finally, the three embedded states are updated as: $z_{\tau+1}^p = z_\tau^c, z_{\tau+1}^r = z_{\tau=0}$, and $z_{\tau+1}^c = z_{\tau+1}^*$.

### 2.3 ACTION SELECTION

The IA repeats the process defined in Section 2.2 of selecting $z_\tau^*$ and $a_\tau^*$ for $T$ steps before finally emitting a final hidden state $h_T^{\mathcal{I}}$ summarizing the result of planning. The OA uses IA's final hidden state $h_T^{\mathcal{I}}$ and its initial hidden state $h_{\tau=0}^{\mathcal{O}}$ to select an action $a_t$:

$$a_t = \boldsymbol{W}^{ah} tanh(\boldsymbol{W}^{hh} h_T^{\mathcal{I}} + h_{\tau=0}^{\mathcal{O}}) \tag{6}$$

where $\boldsymbol{W}^{hh} \in \mathbb{R}^{h^o \times h^i}$ and $\boldsymbol{W}^{ah} \in \mathbb{R}^{\mathcal{A} \times h^o}$ are learnable parameters of OA. Finally, the hidden state of the IA is reset.

### 2.4 TREE INTERPRETATION

The planning process can be interpreted as dynamically expanding a state-action tree, illustrated in Figure 3, where all edges and vertexes are chosen by IA to maximize the total *utility* provided to OA.

With simulated-state selection $z_\tau^*$, using $w_\tau$, IA controls which node in the tree is expanded further: the parent node ($z^p$), the root node ($z^r$), or the current node ($z^c$). While action selection $a^*$ chooses the branching direction, exploring the embedded state space using the state-transition model.

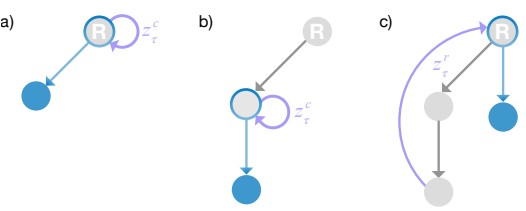

Figure 3: Example of dynamic tree construction during planning.

The illustration of a constructed tree in a fictional environment is shown in Figure 3. State selections are shown in light purple, and state transitions with an action, using the state-transition model, are shown as blue. The source state is shown as a grey circle with a blue outline and the transitioned state is shown as a fully blue circle. In this example, there are three actions, each corresponding to their graphical representation: left, right, and down. The root state is marked with an "R". An example of a possible tree construction for T=3 steps of planning: a) step $\tau = 1$, IA selects the current state $z^c$ and transitions to a new state with action "left"; b) step

$\tau = 2$, the IA selects the current state $z^c$ and "down" action; and step $\tau = 3$ c) IA selects the root state $z^c$ and "down" action.

## 2.5 TRAINING PROCEDURE

Our architecture is trained to maximize the expected rewards within the environment. The OA is trained to maximize the expected discounted sum of rewards, $R_t = \sum_{t=0}^{\infty} \gamma^t r_t$, while the IA maximizes the *utility* it provides to OA. The OA learns a deterministic policy $\pi(a_\tau, z_\tau)$ that directly maps states $s \in \mathcal{S}$ to actions $a \in \mathcal{A}$, while IA learns a stochastic policy $\pi(a_\tau | z_\tau)$. Empirically, we found that using the same policy for planning and acting caused poor performance. We hypothesize that the optimal policy for planning is inherently different from the one required for optimal control in the environment; as during planning, a bias toward exploration might be optimal.

OA uses an identical training procedure and loss to that of DQN used in Mnih et al. (2015) with the loss denoted as $\mathcal{L}_{\mathcal{O}}$. We used the Huber loss to perform state-grounding of the state-transition model between the current state $z_t$, action $a_t$, and $z_{t+1}$ which we denoted with $\mathcal{L}_{\mathcal{Z}}$. IA is trained using a policy gradient method, given in Equation 7, defined over a planning episode $\tau = 1, ...T$:

$$\nabla_{\theta^{\mathcal{I}}} J(\theta^{\mathcal{I}}) = E_{\pi_{\mathcal{I}}}[\nabla_{\theta^{\mathcal{I}}} log \pi_{\mathcal{I}}(z_\tau, a_\tau) \mathcal{R}_\tau] \tag{7}$$

where $\mathcal{R}_\tau$ is defined as the discounted *utility* $\sum_{\tau=1}^{T} \mathcal{U}_\tau + \gamma^{\mathcal{I}} \mathcal{U}_{\tau+1}$ for planning step $\tau$. The IA can be interpreted as an actor-critic algorithm but where the value function is learned by the OA. Combining our losses, the architecture is trained using the following gradient:

$$\Delta \theta = \nabla_\theta \mathcal{L}_{\mathcal{O}} + \lambda \nabla_\theta \mathcal{L}_{\mathcal{Z}} + \frac{1}{T} \sum_{\tau=0}^{T} \nabla_{\theta^{\mathcal{I}}} log \pi_{\mathcal{I}}(a_t | z_t) \mathcal{R}_\tau - \beta_{\mathcal{A}} \nabla_{\theta^{\mathcal{I}}} H[\pi(z_\tau)] - \beta_w \nabla_{\theta^{\mathcal{I}}} H[w_\tau] \tag{8}$$

where $\lambda$ controls the state-grounding loss and $\beta_{\{\mathcal{A}, w\}}$ are hyperparameters tuning entropy maximization of IA's policy. The losses $\mathcal{L}_{\mathcal{O}}$ and $\mathcal{L}_{\mathcal{Z}}$ are computed over all parameters; while the policy gradient and entropy maximization losses are with respect to only IA's parameters. We perform updates to IA in this way as to stop IA from cheating by modifying the parameters of the OA that define its reward via $D_{KL}$ and $Q(z, a; \theta^{\mathcal{O}})$ within the *utility*.

## 3 RELATED WORK

Various efforts have been made to combine model-free and model-based methods, such as the *Dyna-Q* algorithm (Sutton, 1991) that learns a model of the environment and uses this model to train a model-free policy. Originally applied in the discrete setting, Gu et al. (2016) extended Dyna-Q to continuous control. In a similar spirit to the Dyna algorithm, recent work by Ha & Schmidhuber (2018) combined data generated from a pre-trained unsupervised model with evolutionary strategies to train a policy. However, none of the aforementioned algorithms use the learned model to improve the online performance of the policy and instead use the model for offline training. Therefore, the learned models are typically trained with a tangential objective to that of the policy such as a high-dimensional reconstruction. In contrast, our work learns a model in an end-to-end manner, such that the model is optimized for its actual use in planning.

Guez et al. (2018) proposed MCTSnets, an approach for learning to search where they replicate the process used by MCTS. MCTSnets replaces the traditional MCTS components by neural network analogs. The modified procedure evaluates, expands, and back-ups a vector embedding instead of a scalar value. The entire architecture is end-to-end differentiable.

Tamar et al. (2016) trained a model-free agent with an explicit differentiable planning structure, implemented with convolutions, to perform approximate on-the-fly value iteration. As their planning structure relies on convolutions, the range of applicable environments is restricted to those where state-transitions can be expressed spatially.

Pascanu et al. (2017) implemented a model-based architecture comprised of several individually trained components that learn to construct and execute plans. Their work supports varied modes of planning such as 1-step, n-step, and tree-based. The tree-based version is similar to our work but

is only able to use the current or root states. In contrast our work provides the IA with access to the previous state allowing it to undo previous actions. They examine performance Gridworld tasks with single and multi-goal variants but on a limited set of small maps.

Vezhnevets et al. (2016) proposed a method which learns to initialize and update a plan; their work does not use a state-transition model and maps new observations to plan updates.

*Value prediction networks* (VPNs) by Oh et al. (2017), *Predictron* by Silver et al. (2016b), and Farquhar et al. (2017a), an expansion of VPNs, combine learning and planning by training deep networks to plan through iterative rollouts. The *Predictron* predicts values by learning an abstract state-transition function. VPNs constructs a tree of targets used only for action selection. Farquhar et al. (2017a) create an end-to-end differentiable model that constructs trees to improve value estimates during training and acting. Both Oh et al. (2017) and Farquhar et al. (2017a) construct plans using forward-only rollouts by exhaustively expanding each state's actions. In contrast to the aforementioned works, during planning DPN learns to selectively expand actions at each state, with the ability to adjust sub-optimal actions, and uses planning results to improve the policy during both training and acting.

Weber et al. (2017) proposed Imagination Augmented Agents (I2As), an architecture that learns to plan using a separately trained state-transition model. Planning is accomplished by expanding all available actions $\mathcal{A}$ of the initial state and then performing $\mathcal{A}$ rollouts using a tied-policy for a fixed number of steps. In contrast, our work learns the state-transition model end-to-end, uses a separate policy for planning and acting, and is able to dynamically adjust planning rollouts. Additionally, in terms of sample efficiency, I2As require hundreds of millions of steps to converge, with the Sokoban environment taking roughly 800 million steps. Though not directly comparable, our work in the Push environment, a puzzle game very similar to Sokoban, requires an order of magnitude fewer steps, roughly 20 million, before convergence.

Within continuous control learning a state-transition model for planning has been used in various ways. Finn & Levine (2017) demonstrate the usage of a predictive model of raw sensory observations with model-predictive control (MPC) where the model is learned in an entirely self-supervised manner. Srinivas et al. (2018) proposed using an embedded differential network that performs iterative planning through gradient descent over actions to reach a specified target goal state within a goal-directed policy. Henaff et al. (2017) focus on model-based planning in low-dimensional state spaces and extend their method to perform in both discrete and continuous action spaces.

Tishby & Polani (2011) proposed a formulation of the Markov decision process where rewards are traded against control information. The notion of *valuable information* is discussed, which determines the relevance of information by the value it allows the agent to achieve. *Utility* is related to this notion of *valuable information* but instead measures the "gain" the OA can expect from a state-transition. Additionally, our work maximizes *both* the expected reward given by the environment and the *utility* provided to OA while their work focuses solely on maximizing control information and does not couple information value with rewards.

Additional connections between learning environment models, planning and controls, and other methods related to ours were previously discussed by Schmidhuber (2015).

## 4 EXPERIMENTS

We evaluated DPN on a multi-goal Gridworld environment and Push, (Farquhar et al., 2017a) a box-pushing puzzle environment. Push is similar to Sokoban used by Weber et al. (2017) with comparable difficulty. Within our experiments, we evaluated our model performance against either model-free baselines, DQN and A2C, or planning baselines, such as TreeQN and ATreeC. The experiments are designed such that a new scenario is generated across each episode, which ensures that the solution of a single variation cannot be memorized. We are interested in understanding how well our model can adapt to varied scenarios. Additionally, we investigate how planning length $T$ affects model performance using the Push environment, planning patterns that our agent learns in the Push environment, and how the IA's target affects the architecture performance. Full details of the experimental setup and hyperparameters are found in the Appendix B. Unless specified otherwise, each DPN model configuration is averaged over 3 different seeds and is trained for 20 million steps due to limited computational resources.

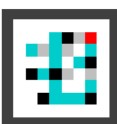 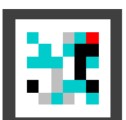 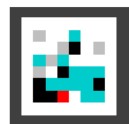 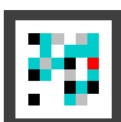 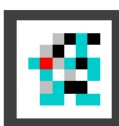 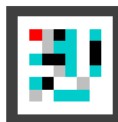

Figure 4: Randomly generated samples of the Push environment. Each square's coloring represents a different entity: the agent is shown as red, boxes as aqua, obstacles as black, and goals as grey. The outside of the environment, not visible to the agent, is shown as a black border around the map.

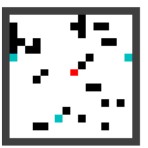 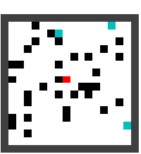 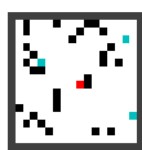 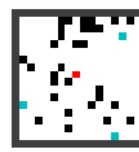 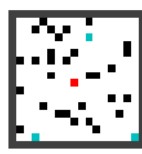 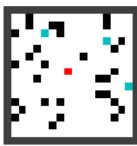

Figure 5: Randomly generated samples of a $16 \times 16$ Gridworld environment where the agent must collect all goals. The agent is shown as red, goals in cyan, obstacles as black, and outside of the environment, not visible to the agent, is shown with a black border.

**Push**: The Push environment is a box-pushing domain, where an agent must push boxes into goals while avoiding obstacles. Samples of this environment are shown in Figure 4. Since the agent can only push boxes, with no pull actions, poor actions within the environment can lead to irreversible configurations. The agent is randomly placed, along with 12 boxes, 5 goals, and 6 obstacles on the center 6x6 tiles of an 8x8 grid. Boxes cannot be pushed into each other and obstacles are "soft" such that they do not block movement, but generate a negative reward if the agent or a box moves onto an obstacle. Boxes are removed once pushed onto a goal. We use the open-source implementation provided by Farquhar et al. (2017b). The reward structure is as follows: +1 for pushing a box onto the goal, -0.2 for moving over an obstacle or pushing a box over an obstacle, -0.1 if the agent attempts to push a box into another, -0.01 for each step, and -1 if the agent falls off the map. A new map is generated at the end of each episode which occurs after 75 steps, if the agent falls off the map, or when all boxes have been cleared. We compare our model performance against planning baselines, TreeQN and ATreeC (Farquhar et al., 2017a), as well as model-free baselines, DQN (Mnih et al., 2015) and A2C (Mnih et al., 2016).

**Planning length**: Using the Push environment, we performed a hyperparameter search over parameter $T$, which adjusts the number of planning steps, with $T = \{1, 3, 5\}$ evaluated. The push environment was chosen because the performance is sensitive to an agent's ability to plan effectively.

**Planning patterns**: We examine the planning patterns that our agent learns in the Push environment for T=3 steps. Here we are interested in understanding what information the agent extracts from the simulation as context before acting.

**Gridworld**: We use a Gridworld domain with randomly placed obstacles that an agent must navigate searching for goals. The environment, randomly generated between episodes, is a 16x16 grid with 3 goals. Details of level generation are provided in the appendix. The agent must learn an optimal policy to solve new unseen maps. Figure 5 shows several instances of a 16x16 Gridworld where the agent is shown as red, goals in blue, and obstacles as black. The rewards that an agent receives are as follows: +1 for each goal captured, -1 for colliding with a wall, -1 for stepping off the map, -0.01 for each step, and -1 for going over the step limit. An episode terminates if the agent collides with an obstacle, collects all the goals, steps off the map, or goes over 70 steps. We evaluate our algorithm against the following variations of the DQN baseline (Mnih et al., 2015): a wide network with double the number of hidden units per layer, a deeper network using an additional hidden layer, and a recurrent version. Each DQN variant used the same encoder structure as DPN.

**Inner Agent Target**: Using the Gridworld environment we examine how the IA target affects the architectures overall performance. To this end we compare the original target, defined in Equation

1, against two variants: the KL divergence between the OAs hidden states and the Q-function target used by the OA.

## 5 RESULTS AND DISCUSSION

### 5.1 PLANNING LENGTH

In Figure 6(a), we see the performance of our model over the planning lengths $T = \{1, 3, 5\}$ where each parameter setting is trained with 3 random seeds for 20 million steps in the environment. As seen in Figure 6(a), model performance increases as we add additional planning steps, while the number of model parameters remains constant.

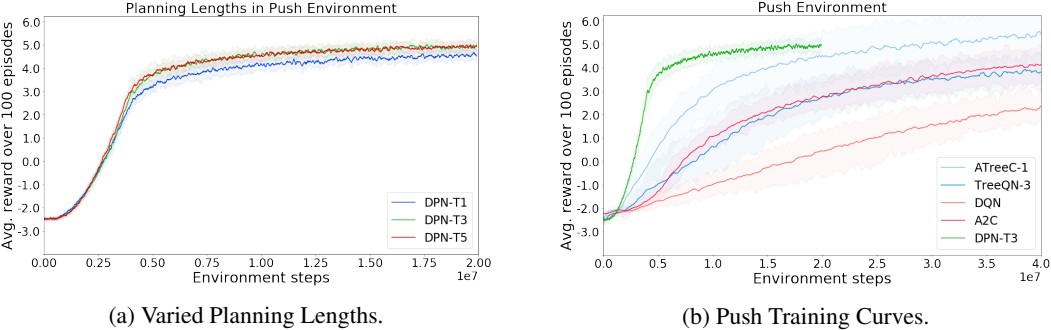

(a) Varied Planning Lengths.      (b) Push Training Curves.

Figure 6: *Push Environment. a)* Training over varying planning lengths, $T = \{1, 3, 5\}$, in the Push Environment. *b)* Training curves of DPN vs. baselines on the Push environment.

As the planning length increases, we see the model converge faster. We begin to see diminishing returns in performance after $T = 3$ planning depth. As seen from the plots, even a single step of planning allows the agent to test action-hypotheses and avoid poor choices in the environment. Ideally, the architecture would be able to adjust the number of planning step $T$ dynamically, similar to the adaptive computation presented by Graves (2016), but we leave this to future work.

### 5.2 PUSH ENVIRONMENT

Figure 6(b) shows DPN, with planning length $T = 3$, compared to DQN, A2C, TreeQN and ATreeC baselines [1]. For TreeQN and ATreeC, we chose tree depths which gave the best performance, corresponding to tree depths of 3 and 1 respectively. Our model clearly outperforms both planning and non-planning baselines: TreeQN, DQN, and A2C; with a slight performance difference to ATreeC. We see that our architecture converges at a much faster rate than the other baselines requiring roughly 12 million steps in the environment, in comparison to the other planning baselines, TreeQN and ATreeC, which take roughly 35-40 million steps: $\sim$3x additional samples.

We note that the planning efficiency of DPN is higher in terms of overall performance per number of state-transitions. On the Push environment, with $\mathcal{A} = 4$ actions, TreeQN with tree depth of $d = 3$ requires $\left(\frac{\mathcal{A}^{d+1}-1}{\mathcal{A}-1}\right) - 1 = 84$ state-transitions. In contrast, DPN with planning length of $T = 3$ requires only $T$ state-transitions – a 96% reduction. Loosely comparing to I2As, simply in terms of state-transitions, we see that I2As require $\mathcal{A} \times L$ state-transitions per action step, where $L$ is the rollout length their model performs. This performance is a result of DPN learning to selectively expand actions and being able to dynamically adjust previously simulated actions during planning.

Additionally, we observed that our model does not suffer from the issue of bouncing between adjacent states; an issue also noted by Farquhar et al. (2017a) with TreeQN. An earlier iteration of our work was affected by this issue as IA employed a deterministic state-action value function. Our solution was to use an actor-critic algorithm for IA to introduce stochasticity into the decision-making

---

[1]The data for the training curves of DQN, A2C, TreeQN, and ATreeC were provided by Farquhar *et al.* via email correspondence. Each experiment was run with 12 different seeds for 40million steps.

processes of OA such that the agent was able to bounce out of these states quickly, thus improving performance.

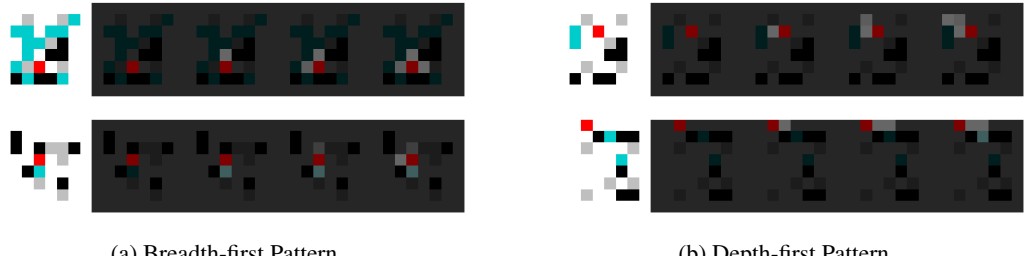

(a) Breadth-first Pattern.                 (b) Depth-first Pattern.

Figure 7: Samples of planning patterns the agent uses to solve the Push environment with $T = 3$. The faded environments, to the right of each sample, is used to signify when the agent is planning. Highlighted squares represent the location that IA chose to move towards during planning.

### 5.3 PLANNING PATTERNS

By watching a trained agent play through newly generated maps, we identified common planning patterns, which are shown in Figure 7. Two prominent patterns emerged: breadth-first search and depth-first search. From Figure 7(a) we can see that our agent learned to employ breadth-first search, where planning steps are used to expand available actions around the agent, corresponding to a tree of depth 1. In contrast, depth-first search as seen in Figure 7(b), has the agent expanding state forward only. The agent does not always follow depth-first search paths and seems to use them to "check" if a particular pathway is worth pursuing.

### 5.4 MULTI-GOAL GRIDWORLD

Figure 8(a) shows, the results of DPN compared to various DQN baselines. Within this domain, the difference in performance is clear: our model outperforms the baselines by a significant margin. The policies that DPN learns generalizes better to new scenarios, can effectively avoid obstacles, and is able to capture multiple goals. Of the DQN variations, we found that DQN-RNN performs better than the other two versions, implying that for a model-free algorithm to perform well within this environment, the agent must be able to perform additional computations and retain some information on previous moves it has made. Additionally, as seen in Figure 8(a), the Deep and Wide DQN variants do not achieve a score higher than -1.0 indicating the agents learn only to navigate around the map without collecting goals before an episode ends. It should be noted we saw little performance improvement with allocating the model-free algorithms an additional 2x environment steps (40 million) or 2-4x longer exploration period (8-16 million).

The poor performance of DQN models is unsurprising as this environment is particularly unforgiving due to: episode termination conditions, goal placement, and density of obstacles. As previously mentioned, episodes end when the agent touches an obstacle, moves off the map, or exceeds the number of allocated steps. While the difficulty with goal placement is that a certain distance must be traversed between the agent to a goal and goal-to-goal, meaning a goal will rarely be discovered without first traversing through several obstacles. Finally, obstacles are often placed in a position where only one square of passage exists, in such positions an incorrect move will cause the episode to terminate leaving little room for error.

### 5.5 INNER AGENT TARGETS

Figure 9(a) shows the performance of the DPN architecture with varied IA targets. From Figure 9(a) we see that the original IA target, labelled Q×KL, converges faster and performs better than both the KL and Q targets. Here, the original loss achieves a +50.49% and +55% greater average performance over the last 1000 episodes as compared to the KL and Q targets respectively.

Interestingly, between the KL and Q targets we see the KL target converges slightly faster and appears more stable throughout training. We hypothesize that when combined together the KL com-

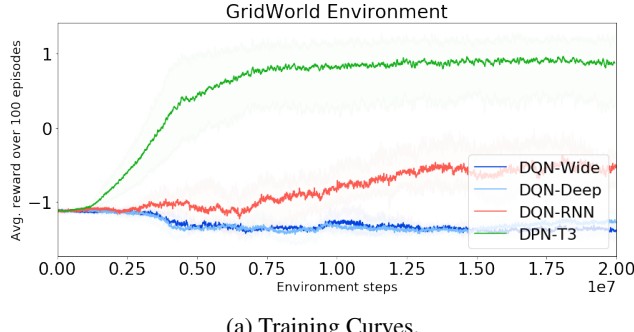

(a) Training Curves.

(b) Model Performance.

Figure 8: *Gridworld Environment. a)* Training curves with DPN compared to various DQN baselines on $16 \times 16$ Gridworld with 3 goals. *b)* The performance of each DQN baseline and our model where *Avg. Reward* is the average of the last 100 episodes of training.

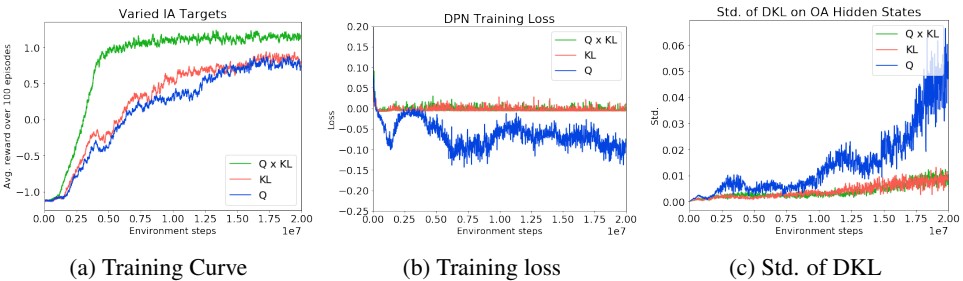

(a) Training Curve

(b) Training loss

(c) Std. of DKL

Figure 9: *Varied IA Targets.* a) *Training curve*: Q×KL corresponds to the original target, KL to the Kullback-Leibler divergence between hidden states of OA, and Q to the Q-function target (as used by the OA). b) *Training loss*: the training loss of the entire architecture. c) *Std. of DKL*: The Std. of the DKL between the hidden states of the OA.

ponent of the original loss acts as a stabilizer to the noisy Q-function helping speed up convergence. We see evidence of this by looking at overall loss in Figure 9(b) and the Std. of the KL between the OA's hidden states in Figure 9(c). The architecture using the Q target experiences greater loss magnitudes and variance throughout training while the KL target has lower variance. The combination of the two producing the original loss, as seen in Figure 9(c), smooths out the variance experienced by the OA.

## 6 CONCLUSION

In this paper, we have presented DPN, a new architecture for deep reinforcement learning that uses two agents, IA and OA, working in tandem. Empirically, we have demonstrated that DPN outperforms the model-free and planning baselines in both the mutli-goal Gridworld and Push environments while using ~3x fewer environment samples. Ablation studies have shown that our proposed target for the IA outperforms other targets and the specific combination helps increase the speed of convergence. We have shown that the IA learns to dynamically construct plans that maximize *utility* for the OA; with IA learning to dynamically use classical search patterns, such as depth-first search, without explicit instruction. Compared to other planning architectures, DPN requires significantly fewer state-transitions during planning for the same level of performance – drastically reducing computational requirements by up to 96%.

ACKNOWLEDGMENTS

*Removed for review as this can leak information about the authors.*

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

# A   SUPPLEMENTAL MATERIAL

## A.1   MOTIVATING EXAMPLE

As this work improves upon existing planning approaches it is pragmatic to examine the general planning process a human might perform before action selection. We use chess as an illustrative example as it is widely known and victory requires careful action selection.

Through introspection the following "planning algorithm" emerges, where before acting each player: a) observes the current game state; b) selects a chess piece to move and mentally steps "forward in time" moving their piece and opponent pieces with a mental dynamics model; and then chooses to ci) repeat step *b)*, cii) undo the last move, ciii) or reset to the current initial "root" state *a)*. With steps ci-iii) repeatedly performed until the player feels they have exhausted all useful paths. Finally, an action is selected using this planning information. This process can be interpreted as dynamically creating a tree structure where the nodes are visited states, the edges are actions, and the tree is grown downward each time the player chooses to step "forward in time". The player traverses this tree by stepping forward *(ci)*, undoing previous moves *(cii)*, or abandoning the current pathway to start from the root state *(ciii)*. Importantly, undoing and resetting does not destroy previous information the player has gained and is remembered when selecting the final action.

From this process we note four interesting characteristics:

- During the planning process, no new external state or information enters the system.
- The entire dynamic process tries to maximize the current information available to the player given only their current state and a dynamics model.
- A model of the environment dynamics is needed to simulate action-conditional steps.
- Previously visited states and transitions should be remembered in working memory instead of being discarded.

Our architecture, DPN, encodes each characteristic into the model structure.

# B   EXPERIMENTAL DETAILS

## B.1   TRAINING DETAILS

We used the RMSProp optimizer with a learning rate of $\alpha = 0.0001$, $\epsilon = 1e - 5$, and decay of 0.95. We trained all environments for 20 million environment steps, using a model freeze interval of 30k, and linearly annealed the exploration rate from 1.0 to 0.05 over the first 4 million steps in the environment. Our replay memory held 1 million samples. We used a discount rate of $\gamma^{\mathcal{O}} = 0.95$ for the outer agent and $\gamma^{\mathcal{I}} = 1.0$ for the IA. The entropy regularization used was $\beta_{\mathcal{A}} = 0.01$ and $\beta_w = 0.007$. The state-grounding coefficient used was $\lambda = 0.01$ in all experiments. All hyperparameters were held fixed during all experiments.

## B.2   GRIDWORLD ENVIRONMENT

For each episode, a new level is generated where we place an agent, 3 goals, and 50 obstacles of varying size with their locations sampled uniformly in a $16 \times 16$ grid map. First, the agent is placed within one of the center 4 tiles of the map. Then a location for each goal is sampled where the location must satisfy the following conditions:

- $D_E[g_i, a] \geq d^{g \to a}$
- $D_E[g_i, g_j] \geq d^{g \to g}$

where $D_E[x, y]$ is the euclidean distance between point $x$ and $y$, $d^{g \to a}$ is the distance between goal $g_i$ and the agent, and $d^{g \to g}$ is the distance between goal $g_i$ and $g_j$. We used $d^{g \to a} = 4.5$ and $d^{g \to g} = 6.0$. Next, we randomly sample locations and dimensions for each obstacle, rejecting already occupied locations, where each obstacle can have a width and height in $\{1, 2\}$. Finally, to ensure that each goal can be reached, we carve a path backwards to the agent remove blocks that stop clear passage.

### B.3 ARCHITECTURES

All encoder layers were separated with ReLU non-linearities unless otherwise specified. Convolution layers are specified with the notation conv-wxh-s-n with *n* filters of size $w \times h$ and stride *s*, and fc-h denotes a fully-connected layer with *h* hidden units.

The state-transition model remained consistent between each environment with only the number of actions $\mathcal{A}$ and embedding size $z$ affecting the number of parameters of the component.

**Gridworld**: The encoder consisted of conv-3x3-1-16, conv-3x3-2-32, conv-4x4-2-32, and fc-128. For the IA's hidden state $h^{\mathcal{I}}$, we used $64$ units and the outer agents hidden state $h^{\mathcal{O}}$ has $64$ units. The hidden state $z$ used $64$ units. The DQN baseline used the same encoder as our architecture as well as the same embedding size $z$.

**Push**: The encoder consists of conv-3x3-1-24, conv-3x3-1-24, conv-4x4-1-48, and fc-128. For the IA's hidden state $h^{\mathcal{I}}$, we used $64$ units and the outer agents hidden state $h^{\mathcal{O}}$ has $64$ units. The hidden state $z$ used $128$ units.

### B.4 KL CALCULATION

To compute the KL divergence between $h^{\mathcal{O}}_{\tau}$ and $h^{\mathcal{O}}_{\tau+1}$, we first apply the inverse tanh function followed by the sigmoid: $\sigma(tanh^{-1}(\cdots))$; the resulting output is interpreted as a joint distribution of $h^o$ independent Bernoulli random variables where the $i$th unit's success probability is given by the units value.

