# OpenReview forum: "Dynamic Planning Networks"
_ICLR.cc/2019/Conference_

### Official Review · AnonReviewer2 · 2018-10-27
**Interesting paper**

**Rating:** 6
**Confidence:** 5

**Review:**

Quality: This paper proposes learning to plan approach that can learn to search with an inner agent; conditioned on the output of the inner agent (IA) the outer agent starts to learn a reactive policy in the environments. The inner agent, different from other searching agent, learns to decide what searching pattern to choose. The presented method shows better computation efficiency than competitive baselines. The main applications are on "box pushing" game and grid-world navigation.

clarity: The paper is well-written.
originality: The paper is original.
significance: This paper shows a promising method to combine traditional search method with machine learning techniques and therefore boost sample-efficiency of RL method.

cons:
1. The dynamics model used to plan is given and fully observable. That means a pure Monte-Carlo tree search can achieve very high accuracy.  In the figure 6, AtreeC can also have good performance after 4e7 steps, even better than the proposed method. I am wondering what would happen if 4e7 steps were applied to the proposed method.
2.  One argument from the paper is that their method is computationally efficient. However, this should be presented in a more realistic test environment. In the push and gridworld environment, 84 steps of planning wouldn't be too bad. So a demonstration of the effectiveness from the proposed method on a visually complex game would be great.

---

> ### Author Response · Authors · 2018-11-25
> **Thank you for your comments**
>
> Thank you for taking the time to review our paper. We appreciate the feedback you have given. We have provided a detail response below.
>
> > The dynamics model used to plan is given and fully observable. That means a pure Monte-Carlo tree search can achieve very high accuracy.
> ==================================
> Yes, we agree that the environment model can be given and is full observable. Can you elaborate on why this is considered a con?
>
> > In the figure 6, AtreeC can also have good performance after 4e7 steps, even better than the proposed method. I am wondering what would happen if 4e7 steps were applied to the proposed method.
> ==================================
> We were tightly constrained by compute and doing several runs over many seeds at this length was unfeasible. As for the overall performance (roughly estimated mean difference is under 10%) we believe an important axis to consider here is the reduction in state-transitions (up to 96%) that gives roughly the same performance. Additionally, if we look at the performance of our model after 20e6 steps we see the ATreeC-1 achieves the same level of performance after ~30e6 steps taking 33% additional steps.
>
> > One argument from the paper is that their method is computationally efficient. However, this should be presented in a more realistic test environment.
> ==================================
> Yes, but this is in terms of state-transitions required which were reduce drastically. The environments we tested are relevant to related work (Push: ATreeC+TreeQN & ~I2A) and Gridworld (Pascanu & many others).
>
>
> > In the push and gridworld environment, 84 steps of planning wouldn't be too bad.
> ==================================
> This is unfeasible as 84 steps of planning would be incredibly slow (the gradient updates). Additionally, as shown in the ablation of the planning lengths we see diminishing returns in the Push environment after 3 steps of planning.
>
>
> > So a demonstration of the effectiveness from the proposed method on a visually complex game would be great.
> ==================================
> We agree that a visually complex environment would be helpful but feel this fits into future work relating to scaling the method up. Additionally, we chose both the Push and Gridworld environments as we could iterate quickly and have reasonable training periods.

---

> > ### Comment · AnonReviewer2 · 2018-12-11
> > **Response**
> >
> > Thank you for your response. Some of my concerns are addressed. I still wonder the  performance after more training steps (4e7). From the figure, even after 4e7 training steps, most of the baseline methods are not converged. I agree the proposed method has improved sample efficiency drastically but there is no proof that the asymptotic performance is acceptable. If you can address this in the final version, I will recommend for acceptance. I adjusted my score accordingly.

---

> > > ### Author Response · Authors · 2018-12-11
> > > **Thanks**
> > >
> > > Thank you for your response and score change. We will work on adding a longer training period to our final version to further address your concerns. It will include 3 seeds of 4e7 iterations on the Push domain.

---

### Official Review · AnonReviewer1 · 2018-10-30
**An interesting proposal of flexible model-based planning**

**Rating:** 6
**Confidence:** 2

**Review:**

I think the ideas proposed in this paper are interesting. The paper is quite clearly written and the authors have provided a thorough reviews of related works and stated how the current work is different. I think this work has some significance for model-based reinforcement learning, as it provided us with a new adaptive way to rollout the simulation. I see the the work as a nice extension/improvement of the I2A (Weber et. al. 2017) and the ATreeC/TreeQN work (Fraquhar et. al. 2017). As the authors pointed out, the L2P agent can adaptive rollout different trajectories by choosing to move back to the root (start state) or move one step backward in the tree (regret last planned action). This is different from ATreeC/TreeQN where the whole tree is expanded in BFS way, and from I2A where rollouts are linear for each possible actions at current state.
I have a bit doubt about the experimental results though. The levels used to evaluate seems quite simple, and I wonder the baseline model-free agents are not properly tuned or are not trained long enough to be fair. I have a list of questions detailed below:

1. The IA is trained with utility that is a measure of "value of information" provided to the OA. I think this is a cool idea. Though I think the readers could understand better the intuition better if the authors can expand the explanation further. any reference on the idea? Why it has the form of Q^ * D_KL, for example why not Q^ + D_KL? Has the authors try to only set the utility as Q^ or as D_KL only as controls?

2. One key part of this model is that during IA's unroll, the agent will choose z* from (z^p, z^c, z^r} (previous, current, root states), and then choose an action to unroll from z*. I wonder if this can be even further extended. For example, one possibility is that the agent can have z* set to any z in the tree that has already expanded. Or, another possibility is that the agent can have z* set to any z along the path from current node to the root (i.e. regret k-steps).  Also, would it be possible to have a dynamic planning steps? These suggestions may be practically hard to work properly, but may worth discuss.

3. "Push is similar to Sokoban used by Weber et. al. (2017) with comparably difficulty". I cannot quite be convinced by this statement. Any quantification or evidence to support this sentence? To me, Sokoban seems to be much harder, as the agent need to solve the whole level to get score and can get stuck if making a single bad decision, while the Push seems much more tolerable (a lot of boxes, the obstacles is softly defined.) So stating that L2P learn Push in an order of magnitude less steps in Push compared to I2A learn Sokoban seems a chicken to egg comparison to me.

4. Is it possible to run I2A as a baseline in the two environment you tried?

5. I don't quite understand why DQN-{Deep, Wide} perform badly in the Gridworld environment. Checking the learning curves, one can see they actually converged to lower score than when the models started (from close to -1 down to -1.3). Can the authors comment more on why this is the case? The authors mentioned 'the agents learn only to navigate around the map for 25-50 steps before an episode ends'. I could not digest this sentence and would hope to understand better. To me, this gridworld level is quite trivial, the agent decide which goal is closest to the agent, and then move forward to that one and then onto next goal sequentially. I would like to understand better why this is a good level to test model-based RL and why model-free RL should have a hard time.

6. a few possible typos:
(1) formula 5, 3rd equation, should it be:
      z*_{tau+1} = z_tau + z'' (double prime instead of single prime)?

(2) The last sentence of the paragraph after equation (5)
     z_{tau+1}^r = z_{tau=0}  (tau+1 instead of tau)

(3) the color indication in Figure 5 caption is wrong. (while the description is fine in the main text)

---

> ### Author Response · Authors · 2018-11-25
> **Thank you for your comments**
>
> Thank you for your thoughtful feedback and suggestions for improvement in our paper. We have addressed your points and incorporated your corrections in the paper. We have provided a detail response below.
>
> > Though I think the readers could understand better the intuition better if the authors can expand the explanation further. any reference on the idea?
> ===========================
> - We have included a short discussion of the target below the equation in the paper.
> - We have provided a section in the supplemental material with a motivating example.
> - No we do not have a reference, the idea evolved from examining how an agent/human might plan ahead and questioning what the core process might be.
>
> >Has the authors try to only set the utility as Q^ or as D_KL only as controls?
> ===========================
> We have included another experiment in the paper that performs an ablation of the IA target and compares between Q * D_KL, Q, and D_KL. We found an interesting result: the KL component helps reduce variance during training.
>
> > I wonder if this can be even further extended.
> ===========================
> We had considered including dynamic planning steps, similar to Graves [1], but we decided to limit the scope of the paper to the core contributions proposed. The other suggestions are quite interesting but we feel the implementation might be difficult and wanted to keep the scope focused. In particular the dynamic set size would be difficult to implement.
>
> We do agree that discussing future would would be helpful and added a short discussion.
>
> [1] - https://arxiv.org/abs/1603.08983
>
>
> > 3. "Push is similar to Sokoban used by Weber et. al. (2017) with comparably difficulty". ===========================
> The Sokoban environment provides dense rewards similar to Push (per step, box onto/off goal, level completion). This can be found in the Appendix D.1 of the I2A paper.
>
> > So stating that L2P learn Push in an order of magnitude less steps in Push compared to I2A learn Sokoban seems a chicken to egg comparison to me.
> ===========================
> In both the Push and Sokoban environments the agent can push boxes into a position that it cannot recover from reducing the score and stopping it from solving the level. Some examples of irreversible moves: pushing 4 boxes together in a square or pushing a box against an edge. You are correct that in Push the obstacles are soft and allow both the agent and boxes to move over them and does not contain single floating obstacles like Sokoban.
>
> Therefore, we feel the environments are still comparable in difficulty.
>
> > 4. Is it possible to run I2A as a baseline in the two environment you tried?
> ===========================
> We decided to omit I2As as we had issues replicating their model and the computational requirements were prohibitive given the hardware available while working on this paper.
>
>
> > Re: poor performance of model-free baselines. Can the authors comment more on why this is the case?
> ===========================
> We believe that the model-free baselines have a hard time with the non-stationary environment (we generate a new map every episode) and the agent does not learn to do much besides how to avoid obstacles (perhaps due to the sparsely distributed goals).
>
> - we performed sanity checks after we saw the poor performance of the model-free baselines by adjusting up the number of unique grids the model-free baselines. We found the model-free baselines all performed well for 1-100 unique maps but performance quickly degraded for a higher number of unique maps. The model-free baselines might have issues handling the huge variance in goal placements, obstacle formations (single boxes, hallways, zig-zag patterns etc. are all possible). Furthermore, we believe that our model performs better because it has a state-transition model that lets it test movement hypothesis and this model captures common structure across all map permutations.
>
> - We believe that the model-free agents only learn to navigate around the environment, avoiding obstacles, before a poor action (deliberate or eps-greedy) causes the episode to terminate. This is why we see the resulting scores for the model-free baselines to be -1 + small number: it gets -0.01 per step and -1 for hitting an obstacle (or going off the map).
>
> To help the model-free baselines we increased training time and exploration budget. We tried to give the model-free agents additional help by doubling the number of training steps from 20million to 40million and by also increasing the exploration duration by 4x (4 million to 16 million frames). Neither avenue increased their scores.
>
> We added comments to our paper explaining the attempted training variations used for the model-free baselines.
>
>
> >6. a few possible typos... + caption colors
> ===========================
> Good catches, we have fixed them. Thank you!

---

### Official Review · AnonReviewer3 · 2018-11-04
**Nice results, but weakened by a mysterious inner objective and lack of novelty**

**Rating:** 4
**Confidence:** 5

**Review:**

This paper proposes a new architecture for model-based deep RL, in which an “inner agent” (IA) takes several planning steps to inform an “outer agent” (OA) which actually acts in the world. The main contributions are to propose a new objective for the IA, and to allow the IA to “undo” its imagined actions. Overall I think this could be a great paper, but it needs further justification of some of the architectural choices and more rigorous analysis/experiments before it will be ready for acceptance.

Pros:
- Nice demonstration of improved data efficiency over existing model-based methods.
- Substantial improvement over other model-based methods in terms of computational cost.
- Interesting qualitative analysis showing discovery of DFS and BFS-like search procedures.

Cons:
- Limited novelty over existing methods.
- It is unclear what in the model contributes to improved performance.

Quality
---------

The results in the paper seem impressive in terms of sample complexity, but I think there needs to be further exploration of the source of the results. I strongly suggest including in a revision a number of ablation experiments to tease these details apart---for example, what do the results look like if the IA uses the same objective as the OA? Does the agent achieve worse performance if it has to restart its imaginations from the root of the tree each time, as is more analogous to MCTS and other previous model-based approaches?

Additionally, there are a few places in the paper where unjustified statements are made. For example, in Section 5.3, the paper states that “we hypothesize that focusing, by repeatedly visiting the same state, the IA ensures that the POI is remembered in its hidden state such that the OA can act accordingly, given this information”. This seems very speculative. It would certainly be very interesting if true, but there needs to be something more than just intuition to back up this hypothesis. I recommend including some probe experiments (e.g., force the IA to take a sequence of such actions, or not, and see what the result on the behavior of the OA is) or removing speculations such as this (or moving it to the appendix).

The literature review is missing some related work, particularly from the realm of model-based continuous control. [1-3] are a few references to start with; these papers take a different approach in that they don’t use tree search but they are still worthwhile discussing. I think a reference to [4] is also missing, which takes a related approach to learning the decisions needed to perform MCTS.

Clarity
--------

Overall, the paper is well-written and I understand what was done and how the architecture works. However, I had a hard time understanding the choice of the inner objective (Equation 1). The paper states that this equation defines the “value of information” and defines it as the product of the KL from the OA’s hidden state prior to the transition to after the transition, multiplied by the Q value estimated by the OA and the action probability of the IA. This is very mysterious to me. Why is this a good objective? Why does the KL term of the hidden state of the OA have anything to do with the value of information? Given that the difference in objective of the IA is one of the main contributions of the paper, this choice needs to be justified, explained, and examined. As mentioned above, it would be best if a revision could include some ablation experiments where the choice of this objective is more closely examined.

More broadly, as mentioned above, it is unclear to me what part of the framework results in improved performance. Is it the ability to “undo” actions (rather than starting over from the root or exhaustively performing BFS), or is it the KL-based reward given to the IA? The paper does not provide any insight into this question, making it unclear what are the key points I should take away.

Minor:
- The colors in the caption of Figure 5 do not match the colors in the figure.
- The colors in Figure 7 are very dark and it is hard to make out what is actually happening in the figure.

Originality
-------------

The objective of the inner agent (Equation 1) appears novel, though as discussed above it is unclear to me what exactly it means and what its implications are. The idea of constructing an imagination tree state-by-state is not particularly novel, and was previously explored by Pascanu et al. (2017). I think this paper deserves more discussion and comparison than it is given in the present work (in particular, compare Figure 3 of the present paper and Figure 2 of Pascanu et al.). In general, the main idea in both papers is the same: have an agent learn to take internal planning steps and construct a planning tree that then informs the final action in the world. The biggest differences from Pascanu et al. are that the present work uses a separate objective for the inner agent, and allows taking a step backwards and returning to the previous state (whereas Pascanu et al. only allowed imagining from the current imagined state or from the root). So, the overall the paper has some new ideas, but is not highly novel compared to previous work. I see the two biggest original contributions as being: (1) the separate objective in the inner agent and (2) the ability for the agent to restart its imagination from the previous imagined state.

Significance
----------------

The results reported by the paper are significant in that they do show dramatic improvement in sample complexity over existing model-free methods, as well as improvement in computational cost over existing model-based methods. However, as discussed above, it is hard for me to know what conclusions I should draw from the paper in terms of what aspects of the approach drive this performance. Thus, I think the lack of clarity in this respect limits the significance of the paper.

[1] Finn, C., & Levine, S. (2017). Deep visual foresight for planning robot motion. In Proceedings of the International Conference on Robotics and Automation (ICRA 2017).
[2] Srinivas, A., Jabri, A., Abbeel, P., Levine, S., & Finn, C. (2018). Universal planning networks. In Proceedings of the 35th International Conference on Machine Learning (ICML 2018).
[3] Henaff, M., Whitney, W., & LeCun, Y. (2018). Model-based planning with discrete and continuous actions. arXiv preprint arXiv:1705.07177
[4] Guez, A., Weber, T., Antonoglou, I., Simonyan, K., Vinyals, O., Wierstra, D., … Silver, D. (2018). Learning to search with MCTSnets. In Proceedings of the 35th International Conference on Machine Learning (ICML 2018).

---

> ### Author Response · Authors · 2018-11-25
> **Thank you for your comments**
>
> We would like to thank the reviewer for their helpful and insightful comments. Our responses to the specific concerns follow. We hope that the changes have improved the paper.
>
> > I strongly suggest including in a revision a number of ablation experiments to tease these details apart.
> ================================
> We agree that this would help improve the clarity of the paper and have included an ablation study of the targets used by the IA. We examined the results over 3 variants: original, Q (same as OA), and KL.
>
> > Does the agent achieve worse performance if it has to restart its imaginations from the root of the tree each time, as is more analogous to MCTS and other previous model-based approaches?
> ================================
> To clarify the IA does not retain hidden state and is reset between OA action steps. The current state the OA is in always acts as the root. We have added a sentence making this explicit in the paper.
>
>
> > Additionally, there are a few places in the paper where unjustified statements are made. For example, in Section 5.3, the paper states that “we hypothesize that focusing, by repeatedly visiting the same state, the IA ensures that the POI is remembered in its hidden state such that the OA can act accordingly, given this information”.
> ================================
> We removed this statement as we did not see a clean way to empirically validate it.
>
>
> > The literature review is missing some related work, particularly from the realm of model-based continuous control.
> ================================
> Thank you for pointing out the missing work. We have added each of the suggested works to our paper.
>
> > However, I had a hard time understanding the choice of the inner objective (Equation 1). The paper states that this equation defines the “value of information” and defines it as the product of the KL from the OA’s hidden state prior to the transition to after the transition, multiplied by the Q value estimated by the OA and the action probability of the IA. This is very mysterious to me. Why is this a good objective? Why does the KL term of the hidden state of the OA have anything to do with the value of information? Given that the difference in objective of the IA is one of the main contributions of the paper, this choice needs to be justified, explained, and examined.
> ================================
> We have included an ablation study that teases apart what the proposed target contributes. We came to an interesting conclusion: the KL component helps stabilize the Q component. Additionally, we added further explanation behind the target after its definition.
>
> > The colors in the caption of Figure 5 do not match the colors in the figure.
> ================================
> Fixed.
>
> > The colors in Figure 7 are very dark and it is hard to make out what is actually happening in the figure.
> ================================
> The intention of darkened images is to show only the planning path in relation to the obscured images. If the planning path (white) and agent (red) are difficult to see we will adjust.
>
> > The idea of constructing an imagination tree state-by-state is not particularly novel, and was previously explored by Pascanu et al. (2017). I think this paper deserves more discussion and comparison than it is given in the present work.
> ================================
> We have expanded the depth of this citation.
>
> > The biggest differences from Pascanu et al. are that the present work uses a separate objective for the inner agent, and allows taking a step backwards and returning to the previous state (whereas Pascanu et al. only allowed imagining from the current imagined state or from the root). So, the overall the paper has some new ideas, but is not highly novel compared to previous work.
> ================================
> We agree that the general idea of creating a dynamic tree-based planning is shared between our work and Pascanu et al. but feel that the following differences are significant:
> - We feel the difficulty of our environments are significantly harder and our work is a step forward in scaling up dynamic tree-based planning architectures. For example within the Gridworld environment: our work uses 16x16 Gridworlds that are randomly generated between episodes ensuring the agent is not able to memorize grid layouts. In contrast Pascanu et al. use a 7x7 gridworld environment with only 4 variants and a perfect environment model.
> - Our work trains all components end-to-end instead of treating each part as a distinct sub graph.
> - We also feel that our work simplifies the architectural complexity for dynamic tree-based planning. We require fewer components, no memory, and allow end-to-end training.

---

> > ### Comment · AnonReviewer3 · 2018-11-29
> > **Response to authors (1/2)**
> >
> > Thank you for the detailed response. I appreciate the additional experiments and clarifications and agree they do improve the paper. However, I still do not feel like I understand all of the architectural choices or their implications. Therefore, I am unfortunately not inclined to change my score.
> >
> > > We agree that this would help improve the clarity of the paper and have included an ablation study of the targets used by the IA. We examined the results over 3 variants: original, Q (same as OA), and KL.
> >
> > Thanks, it is very interesting to see that the KL loss in and of itself is potentially useful.
> >
> > > To clarify the IA does not retain hidden state and is reset between OA action steps. The current state the OA is in always acts as the root. We have added a sentence making this explicit in the paper.
> >
> > Perhaps my question wasn't quite clear---I was referring to the strategy taken by the IA within one OA step. That is, the IA currently has the option of taking an action either from {root, parent, current}. What if this set were different, as in it could only take an action from one of {root, current}? This strategy is more similar to the MCTS search strategy, which either keeps expanding the current node until a leaf is reached or restarts from the root. Similarly, what if the IA could only take an action from {current}? This strategy would be equivalent to performing a single rollout. Or what about just from {root}, which is equivalent to doing several 1-step look aheads? I think these comparisons are important to show in the paper because if the {root, parent, current} strategy is not in fact better than {current} then it suggests the agent hasn't really learned a particularly useful planning strategy (or the environment is not interesting enough). If the {root, parent, current} strategy is not better than {root, current} then it limits the novelty over prior work.
> >
> > > We removed this statement as we did not see a clean way to empirically validate it.
> >
> > Ok.
> >
> > > Thank you for pointing out the missing work. We have added each of the suggested works to our paper.
> >
> > Thanks!
> >
> > > We have included an ablation study that teases apart what the proposed target contributes. We came to an interesting conclusion: the KL component helps stabilize the Q component. Additionally, we added further explanation behind the target after its definition.
> >
> > Thanks for adding the additional clarification. However, I'm unfortunately still not sure I understand the motivation behind the KL component. KL is a measure of the distance (or information gain) between two distributions, but h_t^O is not a distribution---all the environments evaluated are deterministic and fully-observable so there is no need to have a distribution over states. Because of this, it's not clear to me why the transition from one state representation to another (h_t^O --> h_{t+1}^O) should be measured via information gain? It seems to me what the KL term is doing is encouraging the agent to search states that are maximally different from the current state under the agent's representation of the world rather than being about information gain. As such, it would seem that, say, maximizing some other distance metric (e.g. L2) between h_t^O and h_{t+1}^O would work equally well.
> >
> > > The intention of darkened images is to show only the planning path in relation to the obscured images. If the planning path (white) and agent (red) are difficult to see we will adjust.
> >
> > It is very hard for me to see the agent especially, and the darker parts of the planning path are also quite difficult to see.
> >
> > > We have expanded the depth of this citation.
> >
> > Thanks.
> >
> > > We feel the difficulty of our environments are significantly harder and our work is a step forward in scaling up dynamic tree-based planning architectures.
> >
> > I agree the environments in the present paper are more challenging (though the gridworld environment in Pascanu et al. was I believe more for probing the generalization behavior of the planning system, not for pushing the limits of difficulty). But evaluating a similar architecture to one that already exists on a slightly harder problem is unfortunately not on its own that compelling.
> >
> > > Our work trains all components end-to-end instead of treating each part as a distinct sub graph.
> >
> > End-to-end training is not always better, so I'm not really convinced by this statement without empirical support to back it up. Also, given the strong architectural assumptions and the fact that the policy gradient and entropy maximization are only computed with respect to IA, the present approach is technically not end-to-end either.

---

> > ### Comment · AnonReviewer3 · 2018-11-29
> > **Response to authors (2/2)**
> >
> > > We also feel that our work simplifies the architectural complexity for dynamic tree-based planning. We require fewer components, no memory, and allow end-to-end training.
> >
> > Well, DPN has (1) an outer agent (trained via Q learning), (2) a recurrent inner agent (trained with policy gradient), and (3) a dynamics model (trained via supervision). IBP has (1) a manager (trained with policy gradient), (2) a controller+memory (trained with policy gradient), and (3) a dynamics model (trained via supervision). Having a recurrent IA is a bit simpler than the controller+memory setup but roughly equivalent in terms of the general idea.
> >
> > Though, in sketching out these differences explicitly I am realizing that there is an important difference in terms of architectural choice that I didn't fully appreciate before. Specifically, in both DPN and IBP there are three choices that have to be made: (1) which external actions to take in the real environment, (2) which states to search from, and (3) which internal actions to take during search. DPN has separate policies for internal and external actions (with the internal policy simultaneously deciding which actions to try and from which states), while IBP has one policy for both internal and external actions but a separate policy for deciding which states to search from. It's not obvious to me which of these is a priori better---they both have their advantages and disadvantages (e.g., DPN is able to learn an internal policy which is more exploratory than IBP; but IBP more naturally handles dynamic numbers of planning steps than DPN). While this makes me feel slightly more favorable towards the novelty of the approach, it does make me wish there were a clearer comparison to tease apart these differences.

---

### Author Response · Authors · 2018-10-26
**Paper title update**

The paper title has been changed to “Dynamic Planning Networks” as the original name is overly generally.

---

### Author Response · Authors · 2018-11-25
**Summary of paper adjustments**

We would like to thank each reviewer for taking the time to review our paper, providing insightful comments, and the thoughtful questions asked. We feel that our paper has been strengthened as a result and are grateful for this outcome.

For the convenience of the reviewers and AC below we summarize the changes to our paper:
- Changed the name from Learning To Plan to Dynamic Planning Networks
- Added additional support to the "why" of the IA target (Added to Sec 2.1)
- Fixed equation typo (Equation 5)
- Fixed equation typo (End of section 2.2)
- Explicitly stated that the IA's hidden state is reset between steps (end of section 2.3)
- Added the 4 citations suggested by AnonReviewer3
- Expanded citation for Pascanu et al. to include more detail and difference between our technique (suggested by AnonReviewer3)
- Updated the caption of Figure 5 (color mismatch)
- Added an ablation study of the IA's target (changes to end of section 4 and 5) as suggested by AnonReviewer3 & AnonReviewer1.
- Added discussion about future work with dynamic planning lengths at the end of section 5.1 (suggested by AnonReviewer1)
- Removed POI comments in section 5.3 (suggested by AnonReviewer3)
- Added additional discussion attempted training improvements to model-free baselines in Section 5.4 (from AnonReviewer1).


After deadline edit:
- We will adding additional ablation experiments to pull apart the performance gains and motivate the usage of the KL distance (or others). As per reviewer3’s suggestions.
- The experiments on the push environment will be extended to 4e7 steps over 3 seeds instead of 2e7 as per AnonReviewer2’s suggestions.

---

### Meta-Review · Area_Chair1 · 2018-11-05
**Potentially very nice paper with clarity issues**

**Confidence:** 3
**Recommendation:** Reject

**Metareview:**


pros:
- Good quantitative results showing clear improvement over other model-based methods in sample efficiency and computational cost (though see Reviewer 2's concerns about the need for more experiments on computational cost).
- Cool qualitative results showing discovery of BFS and DFS
- Potentially novel approach (see cons)

cons:
- Lack of clarity especially concerning equation (1).  Both Reviewers 1 and 3 were unsure of the rationale for this equation which lies at the heart of the method.  It looks to me like a combination of surprise and value but the motivation is not clear.  There are a number of other such places pointed out by the reviewers where model choices were made that seem ad hoc or not well motivated.
- In general it's hard to understand which factors are important in driving the results you report.  As Reviewer 3 points out, more ablation studies and analysis would help here.  Providing more motivation, explanation and analysis would help the reader understand better the reasons for the performance of the model.

The results are nice and the method is intriguing.  I think this potentially a very nice paper and if you can address the above concerns but isn't quite up to the acceptance bar for ICLR this year.